# A Review of Trastuzumab Biosimilars in Early Breast Cancer and Real World Outcomes of Neoadjuvant MYL-1401O versus Reference Trastuzumab

**Charlie Yang [1,†], Raida Khwaja [2,†], Patricia Tang [1], Nancy Nixon [1], Karen King [2] and Sasha Lupichuk [1,*]**

[1] Department of Oncology, Tom Baker Cancer Centre, Calgary, AB T2N4N2, Canada; charlie.yang@albertahealthservices.ca (C.Y.); patricia.tang@albertahealthservices.ca (P.T.); nancy.a.nixon@albertahealthservices.ca (N.N.)

[2] Department of Oncology, Cross Cancer Institute, Edmonton, AB T6G1Z2, Canada; raida.khwaja@albertahealthservices.ca (R.K.); karen.king3@albertahealthservices.ca (K.K.)

[*] Correspondence: sasha.lupichuk@ahs.ca

[†] Co-first authors.

**Abstract:** The reduced cost of trastuzumab biosimilars has led to increased adoption for HER2-positive breast cancer. This review of trastuzumab biosimilars encompasses this development and real world clinical data in early breast cancer. In addition, we present a retrospective study evaluating the total pathological complete response (tpCR) rates (lack of residual invasive cancer in resected breast tissue and axillary nodes), of MYL-1401O to reference trastuzumab (TRZ) in the neoadjuvant setting for HER2+ early breast cancer (EBC) in Alberta, Canada. Neoadjuvant patients with HER2+ EBC treated with TRZ from November 2018–October 2019 and MYL-1401O from December 2019–September 2020 were identified. Logistic regression was used to control for variables potentially associated with tpCR: trastuzumab product, age, pre-operative T- and N-stage, grade, hormone receptor (HR)-status, HER2-status, chemotherapy regimen, and chemotherapy completion. tpCR was 35.6% in the MYL-1401O group ($n$ = 59) and 40.3% in the TRZ ($n$ = 77) group, $p$ = 0.598. After controlling for clinically relevant variables, there was no significant difference in the odds of achieving tpCR in patients treated with TRZ versus MYL-1401O (OR 1.1, 95% CI 0.5–2.4, $p$ = 0.850). tpCR rates were similar for patients treated with MYL-1401O compared to trastuzumab in our real world study of HER2+ neoadjuvant EBC and comparable to pivotal phase 3 trials.

**Keywords:** trastuzumab; biosimilar; MYL-1401O; HER2+, breast cancer; early stage

## 1. Introduction

Human Epidermal Growth Factor Receptor-2 (HER2) overexpression occurs in 15% to 20% of early stage breast cancers [1]. Clinical trials evaluating the addition of trastuzumab, a monoclonal antibody directed against the extracellular HER2 domain, have demonstrated significant clinical benefits in early and metastatic disease [2,3]. Despite this, fewer than 10% of patients in some low-income countries have access to trastuzumab due to economic restraints [4]. The World Health Organization formally listed trastuzumab in its list of essential medications due to the adverse health implications and years of life lost from lack of access [5]. Trastuzumab biosimilars are a lower cost alternative and have the potential to increase accessibility without compromising clinical efficacy. Given the increasing role of trastuzumab biosimilars in the curative management of HER2 positive breast cancer, we conducted a systematic review of current agents. In addition, we evaluated real world effectiveness of MYL-1401O, a trastuzumab biosimilar compared to reference trastuzumab (TRZ) as measured by total pathological

complete response rates (tpCR) to neoadjuvant chemotherapy patients with early HER2+ breast cancer treated in Alberta, Canada.

The crucial development that increased the availability of HER2 targeted therapy to the broader market was trastuzumab biosimilars. Biosimilars are defined as drugs that contain a version of the reference product's active substance with similar biological characteristics, efficacy, and safety [6]. Unlike small-molecule drugs whereby identical generic versions may be mass-produced, manufacturers often have proprietary rights to the living cell lines and processes involved in producing biologic agents. The first step involves creating a recombinant protein with similar biochemical and tertiary structures, activity, and stability as the reference drug. The biosimilar candidate then undergoes a phase 1 study demonstrating comparable pharmacokinetics (PK) and pharmacodynamics to the reference product [7]. Once a biosimilar candidate has been deemed sufficiently similar in conventional PK parameters as its reference drug, developers can progress onto phase 3 studies where clinical efficacy such as progression-free survival (PFS), overall survival (OS), and safety are explored. Often only one phase 3 trial is required, and subsequently biosimilars can be approved for any indications associated with its reference drug through a process of extrapolation without the need to conduct comparative trials for each indication [8]. This accelerated drug development timeline leads to cost savings. The European Medical Agency (EMA) has additional recommendations for designing phase 3 trials, including ensuring that the study is adequately powered, randomized, double-blinded, and parallel grouped with equivalence trial design for at least one indication. Additionally, both the EMA and Food and Drug Agency (FDA) advocate trialists to design a study with minimal heterogeneity within the study population to increase its sensitivity in order to detect clinically meaningful differences between the biosimilar and its reference drug [8,9].

There are, however, differences between biosimilars and the original product. Minor changes in operation such as pH and temperature, use of different cell lines, downstream purification processes, and storage mean that biosimilars are never identical copies of the original product [10]. These inherent differences in manufacturing and perception of clinically meaningful (and potentially harmful) differences are barriers to the uptake of biosimilars by clinicians.

Several trastuzumab biosimilars and biosimilar candidates have undergone phase 3 trials, including CT-P6 (Herzuma), SB3 (Ontruzant), PF-05280114 (Trazimera), ABP 980 (Kanjinti), MYL-1401O, BCD-022 (HERtiCAD), HD201, EG12014, HX102, and TX05. Five of these are currently FDA/EMA-approved (CT-P6, SB3, PF-05280114, ABP 980, and MYL-1401O) [11–15]. MYL-1401O, BCD-022, and HLX02 have phase 3 studies on metastatic breast cancer exclusively, whereas SB3, ABP 980, HD201, EG12014, and TX05 have phase 3 studies in early breast cancer patients [16–20]. CT-P6 and PF-05280014 conducted phase 3 studies in both MBC and EBC [21–24]. Table 1 summarizes all the phase 3 trials completed to date on various HER2-targeted biosimilars.

**Table 1.** Studies to date of trastuzumab biosimilars.

| Biosimilar [Reference] | Population (N) | Primary Outcome | Primary Outcome Results | PFS |
|---|---|---|---|---|
| SB3 [25] | NA + A (800) | bpCR | SB3 51.7% vs. TRZ 42.0% RR 1.26 (95% CI 1.09–1.46) | - |
| ABP-980 [18] | NA + A (725) | tpCR | ABP-980 48% vs. TRZ 41% RR 1.19 (90% CI 1.03–1.37) | - |
| PF-05280014 [22] | MET (707) | ORR | PF-05280014 62.5% vs. TRZ 66.5% RR: 0.94 (95% CI 0.84–1.05) | PF-05280014 12.16 vs. TRZ 12.06 months RR: 1.00 (95% CI 0.80–1.26) |

| | | | | |
|---|---|---|---|---|
| | NA (226) | Cycle 5 trough > 20 ug/mL | PF-05280014 92.1% vs. TRZ 93.3% | - |
| CT-P6 [24] | NA + A (549) | tpCR | CT-P6 46.8% vs. TRZ 50.4% RR: 0.93 (95% CI 0.78–1.11) | - |
| | MET (475) | ORR | CT-P6 57% vs. TRZ 62% | CT-P6 11.07 vs. TRZ 15.52 months |
| BCD-022 [26] | MET (126) | ORR | BCD-022 49.6 vs. TRZ 43.6% | - |
| MYL-1401O [16] | MET (500) | ORR | MYL-1401O 69.6% vs. TRZ 64.0% RR: 1.09 (90% CI 0.97–1.21) | At 48 weeks: MYL-1401O 44.3% vs. TRZ 44.7% RR: 0.95 (95% CI 0.71–1.25) |
| TX05 [20] | NA (674) | tpCR | TX05 48.8% vs. TRZ 45.3% RR: 1.08 (95% CI 0.92–1.27) | - |
| EG 12014 [19] | NA + A (807) | tpCR | tpCR rates n/a in abstract RR: 0.99 (90% CI 0.88–1.12) | - |
| HLX02 [27] | MET (649) | ORR | HLX02 71.3% vs. TRZ 71.4% | HLX02 11.7 vs. TRZ 10.6 months HR: 0.83 ($p = 0.09$) |
| HD201 [17] | NA + A (502) | tpCR | HD201 49.8% vs. TRZ 51.9% | - |

NA (neoadjuvant), A (adjuvant), MET (metastatic), bpCR (breast pathological complete response), tpCR (total pathological complete response), ORR (overall response rate), TRZ (reference trastuzumab), RR (relative risk), PFS (progression free survival).

Limited real-world information is available for the use of trastuzumab biosimilars outside of clinical trials (Table 2). The first is a Danish population-based study that examined the efficacy of neoadjuvant SB3 in combination with pertuzumab and chemotherapy in early HER2+ breast cancer. Of the 215 patients enrolled, 116 (56%) achieved tpCR. Another study reported treatment outcomes across all stages of HER2+ breast cancer who received concurrent chemotherapy plus pertuzumab with either TRZ or SB3 at a single treatment center. Out of the 78 patients enrolled, 24 patients received neoadjuvant SB3 and 43 patients received neoadjuvant TRZ. tpCR rates were 50% in the SB3 group and 58% in the TRZ group, comparable with landmark neoadjuvant TRZ plus pertuzumab combination studies [28–30]. Bae et al. reported real-world outcomes from a Korean-based population with early-stage or metastatic HER2+ breast cancer. In total, 254 patients had early-stage breast cancer and 103 patients had metastatic disease. Patients were eligible if they had received neoadjuvant pertuzumab, chemotherapy, and either CT-P6 or TRZ. In the neoadjuvant group, the tpCR rate was 74.4% in the CT-P6 group versus 69.8% in the TRZ group, though this was not statistically significant. PFS in the metastatic group were similarly numerically higher in the TRZ group but also not statistically significant (13-month CT-P6 vs 18-month TRZ). Other secondary outcomes in cardiac safety, LVEF decline, ORR, and disease control rate did not show any statistically significant difference between the two groups [31]. Finally, Hester et al. examined the difference in pattern of usage and safety outcomes between ABP 980 and TRZ using historical cohorts before and after the trastuzumab biosimilar became available at four Balverian University Centres. Safety outcome was defined by the tpCR rate after neoadjuvant therapy. Out of 530 patients enrolled, only 79 patients received neoadjuvant treatment. Due to the small sample size, significant fluctuations in tpCR rates were observed, ranging from 33% in the TRZ cohort to 55% in the ABP 980 cohort [32].

**Table 2.** Real world effectiveness of trastuzumab biosimilars in retrospective studies.

| Biosimilar (Reference) | Population (N) | Primary Outcome | Primary Outcome Results |
|---|---|---|---|
| SB3 [33] | NA (215) | tpCR | SB3 56% |
| SB3 [28] | NA (67) | tpCR | SB3 50% vs. TRZ 58% ($p$ = 0.532) |
| CT-P6 [31] | NA (254) | tpCR | CT-P6 74.4% vs. TRZ 69.8% ($p$ = 0.411) |
|  | MET (103) | PFS | CT-P6 13.0 vs TRZ 18.0 months ($p$ = 0.976) |
| ABP 980 [32] | NA (79) | tpCR | ABP 980 55% vs. TRZ 33–55% |
| MYL-1401O [34] | NA (136) | tpCR | MYL-1401O 39% vs. TRZ 40.3% ($p$ = 0.598) |

NA (neoadjuvant), MET (metastatic), tpCR (total pathological complete response), PFS (progression free survival), TRZ (reference trastuzumab).

Preference for uptake of a biosimilar in the real world has also been a matter of interest. For example, a national survey from Brazilian oncologists investigated the comfort level using biosimilars and switching from TRZ. Where the reference biologic was available, 63% of the respondents answered they would use biosimilar in all settings where the reference biologic was approved, 35% would use biosimilar only for the setting it was studied in (i.e., the use of MYL-1401O in only the metastatic setting based on the HERITAGE trial), and 2% would not prescribe biosimilars in any clinical setting. Reasons for not switching included the paucity of evidence that guides switching, such as timing (i.e., when to switch over), dosing and adverse events, in addition to concerns with drug efficacy when biosimilar usage is extrapolated from a different indication [35].

In Alberta, MYL-1401O has fully replaced reference trastuzumab in the treatment of early HER2+ breast cancer since December 2019. As the phase III study that led to the approval for MYL-1401O was in the metastatic setting, we sought to understand its real-world effectiveness for early breast cancer with neoadjuvant chemotherapy. This retrospective study compares the tpCR rates in a contemporary cohort of patients who received neoadjuvant MYL-1401O + chemotherapy with a historical cohort of patients who received neoadjuvant trastuzumab + chemotherapy.

## 2. Materials and Methods

### 2.1. Literature Review

Appraisal of published peer-reviewed journal articles in English from any date to the date of access (15 September 2021) on Pubmed using search terms "biosimilar" and "HER2" and "breast" resulted in 85 articles which were then reviewed by the authors. Additional reference lists of narrative and systematic reviews and included trials were hand-searched for potentially relevant citations.

### 2.2. Study Cohort

All patients in Alberta with HER2+ early breast cancer who received neoadjuvant chemotherapy plus either TRZ from November 2018–October 2019 or MYL-1401O from December 2019–September 2020 were identified from the Cancer Care Alberta Breast Data Mart (BDM). The BDM is a database containing all breast cancer patients diagnosed from 1 January 2004 onwards in Alberta, Canada. The data extracted from this database includes patient demographics, tumor characteristics, surgical intervention, Cancer Control Alberta clinic visits, systemic therapies administered, and clinical information such as weight, height, and vitals. The information is prospectively collected from various sources including the Alberta Cancer Registry (ACR), the Cancer Centre Electronic Medical Record (ARIA MO), the Discharge Abstract Database (DAD), and the National Ambulatory

Care Reporting System (NACRS). Treatment details, pre-chemotherapy clinical stage, and post-operative pathological stage were verified through chart review. The definition of HER2+ disease was 3+ overexpression by immunohistochemistry or HER2 amplification via in situ hybridization (ISH) as per ASCO/CAP guidelines [36]. Patients must have completed neoadjuvant treatment with either TRZ or MYL-1401O (crossover was not allowed) and must have proceeded with surgical resection prior to chart review. Patients with known metastases at diagnosis or identified during the neoadjuvant phase were excluded.

*2.3. Outcome Measures*

The primary endpoint was total pathological complete response (tpCR), defined as the absence of invasive cancer in both resected tissue from the breast and resected axillary nodes.

*2.4. Statistical Analysis*

Characteristics and tpCR rates for the TRZ and MYL-1401O groups were compared using Chi Square, Fisher's Exact, t-test, or Mann-Whitney testing where appropriate. A binary logistic regression model was used to estimate the odds of tpCR for those exposed to TRZ compared with those exposed to MYL-1401O while controlling for variables deemed to be clinically relevant: age (<40 vs. 40+), pre-operative clinical T-stage (T1/2 vs. T3/4), pre-operative clinical nodal status (negative vs positive), grade (I/II vs. III), hormone receptor (HR) status (estrogen receptor (ER) and/or progesterone receptor (PR) positive vs ER/PR negative), HER2 (3+ vs. ISH+), chemotherapy regimen (anthracycline-containing vs not), and chemotherapy completion (yes vs. no). Exposure to pertuzumab was not included as a variable given the small numbers and similar proportions exposed in both groups. Mean time from biopsy to first chemotherapy or surgery were not included as variables as these measurements are not known to impact outcomes in HER2+ breast cancer. Odds ratios (OR) and 95% confidence limits (95%CI) are reported. All statistical tests used in this study were two-sided and the significance level was defined a priori as <0.05. Data were analysed using SPSS 22.0 (IBM Corp., Armonk, NY, USA).

The study was deemed minimal risk and consistent with quality assurance research as per the Alberta Research Ethics Community Consensus Initiative [37].

**3. Results**

In total, 136 patients were included as of our data cut-off of 7 January 2021. Most patients were over 40 years (80.9%) and had tumours that were HR-positive (66.2%), HER2+ by IHC 3+ (90.4%), grade 3 (74.3%), clinical T1/T2 (78.7%) and clinically node-positive (75.0%) (Table 3). In terms of chemotherapy backbone, more patients received a non-anthracycline-based regimen (61% non-anthracycline vs 39.0% anthracycline + taxane). Very few patients received the addition of pertuzumab (3.7%) due to lack of public funding. For the cohorts of interest, 77 patients received TRZ, and 59 patients received MYL-1401O. Overall, these two treatment groups were well-balanced, although patients in the MYL-1401O group had higher rate of clinically node-negative disease (39% vs 14.3% TRZ, *p* = 0.001) and longer mean time from biopsy to chemotherapy (1.5 mo vs. 1.1 mo TRZ, *p* = 0.002).

**Table 3.** Cohort characteristics.

| | Total (*n* = 136) | TRZ (*n* = 77) | MYL-1401O (*n* = 59) | *p*-Value |
|---|---|---|---|---|
| Mean age in years | 50.6 | 51.7 | 49.2 | 0.190 |
| Age < 40 | 26 (19.1%) | 13 (16.9%) | 13 (22.0%) | 0.512 |
| Age 40+ | 110 (80.9%) | 64 (83.1%) | 46 (78.0%) | |
| HR- | 46 (33.8%) | 23 (29.9%) | 23 (39.0%) | 0.266 |
| HR+ | 90 (66.2%) | 54 (70.1%) | 36 (61.0%) | |

| | | | | |
|---|---|---|---|---|
| HER2 3+ | 123 (90.4%) | 67 (87.0%) | 56 (94.9%) | 0.120 |
| ISH+ | 13 (9.6%) | 10 (13.0%) | 3 (5.1%) | |
| Grade 1/2 | 35 (25.7%) | 17 (22.1%) | 18 (30.5%) | 0.265 |
| Grade 3 | 101 (74.3%) | 60 (77.9%) | 41 (69.5%) | |
| Clinical T1/T2 | 107 (78.7%) | 57 (74.0%) | 50 (84.7%) | 0.166 |
| Clinical T3/T4 | 23 (20.6%) | 20 (26.0%) | 9 (15.3%) | |
| Clinical N- | 34 (25.0%) | 11 (14.3%) | 23 (39.0%) | 0.001 |
| Clinical N+ | 102 (75.0%) | 66 (85.7%) | 36 (61.0%) | |
| Mean time from biopsy to 1st chemotherapy (months) | 1.3 | 1.1 | 1.5 | 0.002 |
| Mean time from biopsy to surgery (months) | 6.1 | 6.0 | 6.2 | 0.274 |
| AT | 53 (39.0%) | 30 (39.0%) | 23 (39.0%) | 0.998 |
| TCb | 83 (61.0%) | 47 (61.0%) | 36 (61.0%) | |
| Chemotherapy completed | 114 (83.8%) | 69 (89.6%) | 45 (76.3%) | 0.058 |
| Neoadjuvant pertuzumab | 5 (3.7%) | 3 (3.9%) | 2 (3.4%) | 1.000 |

TRZ (reference trastuzumab), HR (hormone receptor), HER2 (human epidermal growth factor receptor-2), ISH (in situ hybridization), N (node), AT (anthracycline + taxane), TCb (docetaxel + carboplatin).

The tpCR rate was 35.6% for patients treated with MYL-1401O which was similar to 40.3% with TRZ ($p$ = 0.598) (Table 4). Using a binary logistic regression model, we could not identify any significant difference in the odds of achieving tpCR in patients treated with TRZ versus MYL-1401O (OR 1.1, 95% CI 0.5–2.4, $p$ = 0.850) (Table 5). After controlling for the variables selected a priori, only those with HR-negative disease had significantly increased odds of tpCR (OR 2.34 95%CI 1.0–5.4, $p$ = 0.043).

**Table 4.** Univariate analysis for tpCR by trastuzumab product.

| | Total (*n* = 136) | TRZ (*n* = 77) | MYL-1401O (*n* = 59) | *p*-Value |
|---|---|---|---|---|
| tpCR—Yes | 52 (38.2%) | 31 (40.3%) | 21 (35.6%) | 0.598 |
| tpCR—No | 84 (61.8%) | 46 (59.7%) | 38 (64.4%) | |

TRZ (reference trastuzumab), tpCR (total pathological complete response).

**Table 5.** Binary logistic regression for tpCR.

| | OR | 95% CI (Lower) | 95% CI (Upper) | *p*-Value |
|---|---|---|---|---|
| TRZ vs. MYL-1401O | 1.079 | 0.491 | 2.367 | 0.850 |
| Age 40+ years vs less | 1.335 | 0.483 | 3.692 | 0.578 |
| HR- vs. HR+ | 2.359 | 1.029 | 5.411 | 0.043 |
| ISH+ vs. HER2 3+ | 0.971 | 0.248 | 3.796 | 0.966 |
| grade 3 vs. grade 1/2 | 1.971 | 0.744 | 5.221 | 0.172 |
| T1/2 vs. T3/4 | 1.049 | 0.400 | 2.747 | 0.923 |
| clinical N+ vs. N0 | 0.474 | 0.181 | 1.244 | 0.129 |
| AT vs. TCb | 0.716 | 0.320 | 1.604 | 0.417 |
| chemotherapy completed: yes vs. no | 2.122 | 0.675 | 6.675 | 0.198 |

TRZ (reference trastuzumab), HR (hormone receptor), ISH (in situ hybridization), N (node), AT (anthracycline + taxane), TCb (docetaxel + carboplatin), OR (odds ratio).

## 4. Discussion

This is the first retrospective study to compare the efficacy of biosimilar MYL-1401O with TRZ in the neoadjuvant setting with respect to tpCR. We demonstrated that the odds of tpCR were similar in both drugs with the 95% confidence interval crossing unity.

In our study, the odds of achieving tpCR was significantly higher only among those with HR-negative disease. This observation is consistent with several prospective and retrospective studies on trastuzumab-containing neoadjuvant chemotherapy regimens in patients with HER2-positive breast cancer [38–43]. Although HER2-targeted therapy has shown benefits in both early and advanced breast cancer regardless of hormone receptor status, it is becoming increasingly clear that hormone receptor expression is associated with trastuzumab resistance. Vici et al. retrospectively examined 872 patients with triple-positive breast cancer treated with adjuvant chemotherapy with or without trastuzumab and found that tumors expressing both ER and PR in >50% of total tumour cells were associated with poorer relapse-free survival and breast-cancer specific survival [44].

The tpCR rates were also similar regardless of whether patients received an anthracycline or non-anthracycline based regimen. This is highly relevant because of the potential for significant cardiotoxicity in anthracycline-based regimens. When used as monotherapy, trastuzumab has been reported to have a 4% risk of cardiotoxicity; when used in combination with anthracycline and cyclophosphamide, this risk can be as high as 27% [45,46]. Even when an anthracycline and then trastuzumab in combination with a taxane were given sequentially in NSABP B-31, 7.8% of patients were unable to start and 15.5% could not complete treatment with trastuzumab due to cardiotoxicity [47]. In the landmark adjuvant BCIRG-006 trial which randomized 3222 patients with operable HER2-positive breast cancer to either AC-T (doxorubicin/cyclophosphamide followed by docetaxel), AC-TH (doxorubicin/cyclophosphamide followed by docetaxel/trastuzumab), or TCbH (doxetaxel/carboplatin/trastuzumab), 10-year DFS and OS were numerically very similar in the two trastuzumab containing arms. Significantly higher rates of grade 3+ congestive heart failure were seen in patients who received anthracycline (2% vs 0.4%) [2]. Prospective studies have also demonstrated similar tpCR rates for anthracycline and non-anthracycline based neoadjuvant regimens. Both the phase 2 TRYPHAENA trial and the phase 3 TRAIN-2 trial have demonstrated comparable tpCR rates and comparable longer term outcomes between the anthracycline and non-anthracycline arms [29,48].

Trastuzumab biosimilars have the potential for significant cost savings to the health care system. They are less costly because they undergo a different approval pathway that does not require the same stages of clinical trials that biologics do—instead, the focus is on ensuring that the biosimilar can achieve an equivalent efficacy and safety profile [49]. A cost-utility analysis by Hoffman-La Roche Limited for the pan-Canadian Oncology Drug Group (pCODR) highlights the monthly cost for TRZ based on listing price and its schedule of administration. In the adjuvant setting, with a mean duration of treatment of 12 months, the cost per patient was $499,916. In the palliative setting, the mean duration was 7.2 months and cost per patient was $28,350. Ducker et al. investigated the economic implication of adopting TRZ to treat breast cancer in Canada. They estimated that the lifetime cost of TRZ is 127 million dollars annually for both early and metastatic HER2-positive breast cancer patients based on 2005 data [50]. In Alberta, the price for TRZ is $2,874.05 per 440 mg vial compared to MYL-1401O, which is $1417.21 per 440 mg vial. Switching to MYL-1401O would potentially lead to significant savings of up to 50.7% per patient. These findings are consistent with the financial impact of switching from TRZ to other biosimilars in Europe. Lee et al. conducted a budget analysis of biosimilar CT-P6 across 28 European countries. Based on their budget impact model, the estimated saving in the first year ranged between 59–136 million euros. Over a 5-year period, this amount would increase exponentially to an estimated 1.13 to 2.27 billion euros based on the assumption that CT-P6 is 70% of the originator price and the switch rate from originator to CT-P6 is 20% in the first year, and 5% annual growth for each subsequent year [51].

Our study has several strengths and limitations. One strength is the non-selected population which captured all patients who were initiated on neoadjuvant HER2-targetted agents in Alberta within the pre-specified timeframe which is more representative of true patient population than the highly selected nature of those enrolled into clinical trials. Another strength is that this is the first real world study comparing MYL-1401O with TRZ in the neoadjuvant setting. Limitations of this study include the retrospective design and small patient sample size. Our study also does not present toxicity data and thus is unable to report real world safety outcomes. Lastly, we could not capture other long-term outcomes such as event-free survival or overall survival, owing to the short follow-up periods.

## 5. Conclusions

Breast cancer research is constantly changing our knowledge of HER2-positive cancers, with newer treatments and targets being proposed annually at an astonishing pace. Despite these advances, the reality is that these will be too expensive for many countries to afford. Biosimilars offer more affordable alternatives, thus potentially granting many more patients access to life-prolonging treatments. Within Alberta, the real-world data analysis of MYL-1401O use in early-stage HER2-positive breast cancer demonstrates similar pathological response rates as TRZ, thus justifying its use within our public health system as an excellent cost-effective alternative to trastuzumab.

**Author Contributions:** Conceptualization, P.T. and S.L.; data curation, C.Y., R.K. and S.L.; methodology, P.T., N.N., K.K. and S.L.; formal analysis, S.L.; supervision, P.T., N.N., K.K. and S.L.; writing—original draft, C.Y. and R.K.; writing—review & editing, C.Y., R.K., P.T., N.N., K.K. and S.L. All authors have read and agreed to the published version of the manuscript.

**Funding:** This research received no external funding.

**Institutional Review Board Statement:** The study was deemed minimal risk and consistent with quality assurance research as per the Alberta Research Ethics Community Consensus Initiative.

**Informed Consent Statement:** Patient consent was waived due to the fact the study was a population-based retrospective analysis.

**Data Availability Statement:** The data that support the findings of this study are available on request from the corresponding author. The data are not publicly available due to privacy or ethical restrictions.

**Conflicts of Interest:** The authors declare no conflict of interest.

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
