# Peer review of "A Review of Trastuzumab Biosimilars in Early Breast Cancer and Real World Outcomes of Neoadjuvant MYL-1401O versus Reference Trastuzumab"

_curroncol, doi:10.3390/curroncol29060337_

Round 1
Reviewer 1 Report
This paper encompasses a literature review of existing evidence on trastuzumab biosimilars, through a review of currently published phase 3 studies and real-world clinical data and a retrospective study evaluating pathological complete response rates of MYL-1401O compared to Trastuzumab in the neoadjuvant setting.
The study is interesting, since further clinical studies on the use of trastuzumab biosimilars in real-life settings are needed to consolidate their use in clinical practice as an alternative to trastuzumab.
I would like to suggest some corrections and ask for some clarification.
Abstract: the abstract is quite clear, but it is very brief leaving a few ambiguous points, such as what is meant by the pathological complete total response.
Introduction:
-
Line 26: remove + after HER2, since the acronym refers to the receptor and not to its positivity.
-
Line 56: check text formatting
-
Table 1 is quite clear but lacks the legend of all acronyms used.
Materials and Methods
-
Materials and methods of the retrospective study are made explicit, while materials and methods of the literature review part are missing.
-
Pay attention to acronyms such as HR, ER, and PGR that have never been used extensively in the text.
Results
-
The use of double blocking in some patients was not analyzed as a confounding factor in the statistical analysis.
-
Tables 2, 3, and 4 are missing the acronym legend. I suggest boldening significant p's and graphically dividing the table into sections (e.g., with thicker lines) to make the table clearer.
Discussion
Discussions are loosely organized.
-
Lines 137 - 140 set out the results of the retrospective study, but then from line 41 to line 179 a part of the literature review is taken up. I suggest moving the literature review section about the use of trastuzumab biosimilars outside of clinical trials after table 1, in the introduction. Combining the entire literature review information into one section would make the text clearer and more effective.
-
Lines 201-244 should be moved after lines 137-140, thus discussing in a more organic and consequential way the results of the prospective study.
-
I would suggest moving the cost-saving effectiveness (lines 180-200) after the discussion of the results of the prospective study and before the strength and limitations of the study.
Line 210 – 219: it appears that this elaboration on in vitro studies of lapatinib-resistant cells and ESR1 gene expression deviates from the objective of the study. I would remove this passage that appears to be overly in-depth, thus superfluous.
I would suggest mentioning the studies proving the cardiotoxicity of trastuzumab before discussing chemotherapy regimens and their side effects.
It would have been interesting to evaluate cardiotoxicity between the two drugs (Trastuzumab vs MYL-1401O)
Conclusions: conclusions are clear.
References: references n° 19, 20, 26 are incomplete, reference 29 must follow the same citation method.
Author Response
Thank you for your queries and suggestions. Please find our responses as outlined below:
Abstract: the abstract is quite clear, but it is very brief leaving a few ambiguous points, such as what is meant by the pathological complete total response.
Response: We have used total pathological complete response (tpCR) to define lack of residual invasive cancer in resected breast tissue AND axillary nodes. This definition has been added to the abstract for clarification.
Introduction:
- Line 26: remove + after HER2, since the acronym refers to the receptor and not to its positivity.
Response: Agree, changed from HER2+ to HER2. Now line 29.
- Line 56: check text formatting
Response: Agree. Period added after reference. Changed format/font to be consistent.
- Table 1 is quite clear but lacks the legend of all acronyms used.
Response: Agree. Legend has a been added. This table has also been significantly streamlined to reflect suggestions from reviewer 3.
Materials and Methods
- Materials and methods of the retrospective study are made explicit, while materials and methods of the literature review part are missing.
Response: Agree. The methods section now has sub-headings. A paragraph has been included now in the methods section for the literature review.
- Pay attention to acronyms such as HR, ER, and PGR that have never been used extensively in the text.
Response: Agree. Clarified acronyms.
Results
- The use of double blocking in some patients was not analyzed as a confounding factor in the statistical analysis.
- Tables 2, 3, and 4 are missing the acronym legend. I suggest boldening significant p's and graphically dividing the table into sections (e.g., with thicker lines) to make the table clearer.
Response: Given the small number of patients who received pertuzumab and the similar proportions per group (3.9% in the TRZ group and 3.4% in the MYL-1401O group), we did not include as a variable in the binary logistic regression. If the proportions had differed by group, this would have been very relevant. Table legends added and table lines made clearer with significant p’s boldened where appropriate.
Discussion
Discussions are loosely organized.
- Lines 137 - 140 set out the results of the retrospective study, but then from line 41 to line 179 a part of the literature review is taken up. I suggest moving the literature review section about the use of trastuzumab biosimilars outside of clinical trials after table 1, in the introduction. Combining the entire literature review information into one section would make the text clearer and more effective.
Response: Agree. Editing and rearrangements of the paragraphs have been completed.
- Lines 201-244 should be moved after lines 137-140, thus discussing in a more organic and consequential way the results of the prospective study.
Response: Agree. Rearrangements of the paragraphs have been completed.
- I would suggest moving the cost-saving effectiveness (lines 180-200) after the discussion of the results of the prospective study and before the strength and limitations of the study.
Response: Agree. Rearrangements of the paragraphs have been completed.
- Line 210 – 219: it appears that this elaboration on in vitro studies of lapatinib-resistant cells and ESR1 gene expression deviates from the objective of the study. I would remove this passage that appears to be overly in-depth, thus superfluous.
Response: Agree. This discussion has been removed.
- I would suggest mentioning the studies proving the cardiotoxicity of trastuzumab before discussing chemotherapy regimens and their side effects.
Response: Agree. We have added a paragraph re: trastuzumab associated cardiotoxicity in this version.
- It would have been interesting to evaluate cardiotoxicity between the two drugs (Trastuzumab vs MYL-1401O)
Response: This is something that we would like to report on with longer term follow-up (i.e. after the patients have completed the year of trastuzumab product). We did not note any neoadjuvant cardiac events in this small cohort.
Conclusions: conclusions are clear.
References:
- references n° 19, 20, 26 are incomplete, reference 29 must follow the same citation method.
Response: Agree. References have been re-inserted.
Reviewer 2 Report
In present paper the influence of the trastuzumab biosimilars was assessed. Authors conducted a systematic review of current agents of trastuzumab biosimilars in the curative management of HER2 positive breast cancer. Also evaluated real world effectiveness of MYL-1401O, a trastuzumab biosimilar compared to reference trastuzumab. The data presented in this paper are very interesting and innovative. The manuscript is prepared very correctly and carefully. Introduction. In my opinion, it is very important part of this section and very well effectuated us to presented in this manuscript problem. The section Methods was well and clearly defined. Authors were used many adequate methods to solve a problem specified for the objective. The section "Results" is very precisely described. Results are well documented. Achieved outcomes are discussed with recent literature. The manuscript can be accepted for further publishing.
Author Response
Many thanks for the supportive review. We feel that it is important to communicate real world data on biosimilars for indications that differ from the one studied in the pivotal trial that brought it to market, especially if the question revolves around curative-intent treatment. We hope that our findings are reassuring for others who have adopted MYL-1401O in the neoadjuvant setting.
Reviewer 3 Report
Dear Prof. Sasha,
The systemic review has been submitted for the review. I went through the content of the review and i found following point of concern.
1. the reference numbers should always put at the end of sentence before the full stop so in all sentences with reference number, this should be followed.
2. Table No. 1 seemed tot be very disorganized and it need to be re construct. it is difficult to understand the relevance of the information within same row.
3. The reference list is quite disorganized without following the format of journal. has to be re write in appropriate manner.
4.Tthe statistical analysis should be explained in separate paragraph and the data significance has to be clearly explained.
I would like to suggest the resubmission of article after removing these basic errors.
Best regards,
Saima
Author Response
We appreciate your time, comments and suggestions. Our responses are inserted below:
- the reference numbers should always put at the end of sentence before the full stop so in all sentences with reference number, this should be followed.
Response: Agree and this has now been corrected.
- Table No. 1 seemed tot be very disorganized and it need to be re construct. it is difficult to understand the relevance of the information within same row.
Response: Agree. This table has been significantly streamlined to highlight relevant information and remove extraneous information.
- The reference list is quite disorganized without following the format of journal. has to be re write in appropriate manner.
Response: Agree. The reference list has been converted to MDPI. A few references were manually corrected (i.e. there were citations which Endnote had titled “…” which I had to re-insert manually).
- The statistical analysis should be explained in separate paragraph and the data significance has to be clearly explained.
Response: The Materials and Methods section now has sub-headings, including statistical analysis. Details of the statistical analysis plan have been expanded. We have attempted to address the data significance throughout the discussion.
Round 2
Reviewer 1 Report
The paper is now fine